# Pravastatin Prevents Increases in Activity of Metalloproteinase-2 and Oxidative Stress, and Enhances Endothelium-Derived Nitric Oxide-Dependent Vasodilation in Gestational Hypertension

**DOI:** 10.3390/antiox12040939

**Published:** 2023-04-16

**Authors:** Cristal Jesus Toghi, Laisla Zanetoni Martins, Leonardo Lopes Pacheco, Edileia Souza Paula Caetano, Bruna Rahal Mattos, Elen Rizzi, Carlos Alan Dias-Junior

**Affiliations:** 1Department of Biophysics and Pharmacology, Institute of Biosciences, Sao Paulo State University (UNESP), Botucatu 18618-689, SP, Brazil; 2Unit of Biotechnology, University of Ribeirao Preto (UNAERP), Ribeirao Preto 14096-900, SP, Brazil

**Keywords:** gestational hypertension, metalloproteinases, antioxidant, pravastatin

## Abstract

Pre-eclampsia (PE) is a hypertensive disorder of pregnancy and has been associated with placental growth restriction. The pre-eclamptic placenta releases free radicals to maternal circulation, thus increasing oxidative stress. An impaired redox state leads to reduction in circulating nitric oxide (NO) levels and activation of extracellular matrix metalloproteinases (MMPs). However, activation of MMPs induced by oxidative stress is still unclear in PE. Antioxidant effects have been demonstrated with the use of pravastatin. Therefore, we hypothesized that pravastatin protects against oxidative stress-induced activation of MMPs in a rat model of PE. The animals were divided into four groups: normotensive pregnant rats (Norm-Preg); pregnant rats treated with pravastatin (Norm-Preg + Prava); hypertensive pregnant rats (HTN-Preg); and hypertensive pregnant rats treated with pravastatin (HTN-Preg + Prava). The deoxycorticosterone acetate (DOCA) and sodium chloride (DOCA-salt) model was used to induce hypertension in pregnancy. Blood pressure, and fetal and placental parameters were recorded. The gelatinolytic activity of MMPs, NO metabolites and lipid peroxide levels were also determined. Endothelium function was also examined. Pravastatin attenuated maternal hypertension, prevented placental weight loss, increased NO metabolites, inhibited increases in lipid peroxide levels, and reduced the activity of MMP-2, and these effects were observed along with enhanced endothelium-derived NO-dependent vasodilation. The present results provide evidence that pravastatin protects against activation of MMP-2 induced by oxidative stress in pre-eclamptic rats. These findings may also involve improvement in endothelial function related to NO and antihypertensive effects of pravastatin, thus suggesting pravastatin as a therapeutic intervention for PE.

## 1. Introduction

Statins are known to inhibit the synthesis of endogenous cholesterol, but recent research has demonstrated other potential benefits [1]. Pleiotropic effects associated with statins include angiogenesis improvement, inhibition of inflammatory response, stabilization of atherosclerotic plaques, antioxidants, enhancement in endothelial function, and increases in nitric oxide (NO) bioavailability [1]. It has been demonstrated that constitutive activity of endothelial NO synthase is directly enhanced by statins, which underlying mechanisms may involve a reduction in caveolin-1 and concomitant increase in heat shock protein 90, thus facilitating the long-term activation of endothelial NO synthase, and thereby increasing endogenous NO synthesis [1].

Pravastatin is a promising and increasingly researched statin that has also shown potential for the prevention of pre-eclampsia (PE) [2,3]. Pleiotropic effects of pravastatin were also experimentally observed, in which attenuation of hypertension, and fetal and placental growth restrictions, were found in a rat model of PE [4,5].

PE incidence in developing countries is 1.8–16.7%, while about 0.4% of incidence is reported in developed countries [6]. Development of PE has shown influences that include environmental and genetic factors, and lifestyle [7]. Although little information is known relating to nutrition/diet and PE, it has been reported that 10% of pregnant obese women develop PE and that these two conditions share common pathophysiological mechanisms [8]; therefore, nutritional guidelines to reduce obesity in pregnancy are of particular importance to attenuate the risk of PE [9]. Thus, PE is a multisystem disease of poorly understood etiology, characterized by endothelial dysfunction, angiogenic imbalance, and exacerbated inflammation [7,10]. Clinical manifestations of PE include elevated blood pressure during pregnancy, proteinuria, and target organ damage [3,7,10]. Although the mechanisms responsible for PE are still unclear, there is a theory that may partially explain its pathogenesis, which is based on inadequate trophoblastic invasion and incomplete spiral artery remodeling, resulting in high-resistance vessels, increases in antiangiogenic factors and oxidative stress, leading to ischemic damage to the placenta [7,10].

The hypoxic environment in a pre-eclamptic placenta results in the release of reactive oxygen species (ROS) and endothelial dysfunction mediators, such as lipid peroxides and pro-inflammatory cytokines, which impair NO bioavailability [10,11,12]. Accordingly, PE has also been associated with decreases in NO bioavailability, which may underlie pathophysiological mechanisms of high vascular resistance of spiral arteries in the pre-eclamptic placenta [13,14,15]. Importantly, earlier results also suggested that NO modulates the activity of matrix metalloproteinases (MMPs) and that may be fundamental in vascular, uterine, and placental changes that occur during pregnancy [16,17,18].

The MMPs are proteolytic zinc-dependent enzymes expressed in various tissues, including smooth muscle, endothelial, fibroblast, and inflammatory cells [19]. Excessive degradation of extracellular matrix components induced by increases in activity of MMP-2 has been associated with hypertensive vascular remodeling, leading to wall thickening, endothelial dysfunction, hyperplasia and hypertrophy of vascular smooth muscle cells, cell migration, proliferation, and apoptosis, which occur in cardiovascular diseases, including PE [20,21]. Previous studies have shown that during hypertensive pregnancy there were increases in MMP-2 in the placenta of rats [16] and in the plasma of women [20] with PE. Indeed, MMP-2 may possibly be influencing the degradation of the placental extracellular matrix, contributing to disturbed placental vascularization and leading to placental growth restriction [16,17]. Moreover, oxidative stress is increased in PE [22], and it may also activate MMPs [23,24], but little is known about whether activity of MMPs may also be affected by oxidative stress in pregnancy with PE.

Therefore, the aim of the present study was to examine the effects of pravastatin on hypertension associated with placental weight reduction induced experimentally in pregnancy, using a well-characterized animal model of PE [4,14]. Our hypothesis is that pravastatin attenuates the increases in activity of MMPs induced by increased oxidative stress in hypertensive pregnancy.

## 2. Materials and Methods

### 2.1. Animals and Experimental Protocols

Female Wistar rats were housed in the bioterium of the Pharmacology Department of the Botucatu Institute of Biosciences, UNESP. The animals were allocated in cages with a 12 h light/dark cycle and controlled temperature (23 ± 2 °C), with access to food and water ad libitum. For mating overnight, the animals were kept in cages in the ratio of two females to one male (*Harem* system) in late afternoon. The following day, the detection of sperm and estrus cells in a vaginal smear confirmed the first day of gestation, and pregnant rats were distributed into four experimental groups:Normotensive Pregnant rats (Norm-Preg group): saline (0.9% NaCl) solution (0.3–0.45 mL) was intraperitoneally (i.p.) administered on days 1, 7, and 14, and saline was administered by gavage from pregnancy day 10 until 19 (n = 8).Normotensive pregnant rats treated with pravastatin (Norm-Preg + Prava group): saline was i.p. administered on days 1, 7, and 14, and pravastatin (10 mg/kg/day) was administrated by gavage from pregnancy day 10 until 19 (n = 8).Hypertensive pregnant rats (HTN-Preg group): hypertension was induced by i.p. administration of 12.5 mg of DOCA on the first day of pregnancy, followed by i.p. injection of 6.5 mg of DOCA on days 7 and 14 of pregnancy; drinking water was replaced by saline from pregnancy day 1 until 19; and saline was administered by gavage from pregnancy day 10 until 19 (n = 8).Hypertensive pregnant rats treated with pravastatin (HTN-Preg + Prava group): hypertension was induced by i.p. administration of 12.5 mg of DOCA on the first day of pregnancy, followed by i.p. injection of 6.5 mg of DOCA on days 7 and 14 of pregnancy; drinking water was replaced by saline from pregnancy day 1 until 19; and pravastatin (10 mg/kg/day) was administrated by gavage from pregnancy day 10 until 19 (n = 8).


On pregnancy day 19, rats were euthanized by overdose of isoflurane followed by exsanguination. Subsequently, a laparotomy was performed for the exposure/removal of the pregnant uterus, and the abdominal aorta was withdrawn. The abdominal aorta was prepared for vascular reactivity experiments. Placental weight and litter size (total number of pups) were recorded. Placenta and plasma were stored at −80 °C until use for biochemical analysis.

### 2.2. Institutional Review Board Statement

All proceedings for experiments in animals were approved by the Institutional Animal Care and Use Committee (protocol n° 3902030620, 14 July 2020) of the Biosciences Institute of Botucatu, Sao Paulo State University. The experimental protocol is in accordance with the European Community for the use of experimental animals’ policy, and it complies with ARRIVE guidelines.

### 2.3. Blood Pressure Measurements

All measurements of systolic blood pressure were made 3 h before any i.p. (drug or saline) injection. Briefly, rats were restrained in acrylic tubes in a quiet room. Then, each animal was contained in a heated box (Insight, Ribeirao Preto, Sao Paulo, Brazil, catalog #EFF-307) for 10 min at 40 °C, and systolic blood pressure was recorded in triplicate by tail-cuff plethysmography (Insight, Ribeirao Preto, Sao Paulo, Brazil, catalog #EFF-306) on gestational day 19, in accordance with the methodology previously described [4].

### 2.4. Vascular Reactivity

Abdominal aorta segments were dissected and cut into four rings (3 mm), in which two rings had their endothelium mechanically removed and two had their endothelium preserved. Each aortic ring was hung between two wire hooks, and placed into an organ chamber containing Krebs–Henseleit solution (NaCl 130; KCl 4.7; CaCl_2_ 1.6; KH_2_PO_4_ 1.2; MgSO_4_ 1.2; NaHCO_3_ 15; glucose 11.1; in mmol/L) kept at pH 7.4 and 37 °C, and bubbled with 95% O_2_ and 5% CO_2_, and then were stabilized under basal tension of 1.5 g [4].

Following aortic rings’ equilibration, KCl maximum contraction was obtained through the administration of KCl (96 mM) to test aorta viability. To examine endothelial function, aortic rings were precontracted with 10^−6^ M of phenylephrine (Phe), and increasing concentrations (10^−9^ to 10^−4^ M) of acetylcholine (ACh) were added [4]. In order to confirm the involvement of endothelium-derived NO-dependent vasodilation, a concentration–response curve to ACh was obtained in the presence of Nω-nitro-L-arginine-methyl ester (L-NAME, 3 × 10^−4^ M) in an aortic ring pre-contracted with Phe. Non-linear regression (variable slope) of the obtained concentration–effect curves revealed the R_max_ (maximal response) and the pEC_50_ (negative logarithm of the concentration that evoked 50% of the maximal response). The relaxation curves were expressed as the % relaxation to Phe-induced contraction, as previously described [4].

### 2.5. Determination of NO Metabolites (Nitrite/Nitrate: NOx) Levels in Plasma

Griess reagents were used to determine NOx, followed by reduction in nitrous species with vanadium chloride III. Briefly, a plasma sample was incubated with 100 μL of saturated solution of vanadium chloride III (for 3 h) at 37 °C. After incubation, 50 μL of sulphanilamide (1%) and phosphoric acid (5%) solution was added, and microplate was incubated for 10 min. Then, 50 μL of N-(1-naphthyl)-ethylenediamine dihydrochloride (0.1%) solution was added followed by 10 min incubation in the dark. Absorbance was read in a spectrophotometer (SYNERGY 4; BIOTEK, Winooski, VT, USA) at a wavelength of 535 nm. The NOx concentration was calculated using a standard curve of sodium nitrite (1.56–100 μmol/L). The plasma NOx concentration was expressed in μmol/L, as previously described [4,16].

### 2.6. Determination of Lipid Peroxide Levels in Plasma

Products of lipid peroxidation were assessed by the thiobarbituric acid (TBA) reactive substances (TBARS) method, which detects the levels of malondialdehyde (MDA), the main product of lipid peroxidation. Briefly, 100 µL of plasma was added into testing tubes and incubated with 100 µL of distilled water, 50 µL of 8.1% sodium dodecyl sulfate (SDS), 375 µL of acetic acid 20%, and 375 µL of TBA 0.8% for one hour in a water-bath at 95 °C. Then, the samples were centrifuged at 4000 rpm for 10 min [4,13]. TBA was added to samples and a colorimetric reaction immediately obtained, which was measured through a wavelength of 532 nm, as previously described [4,16]. The plasmatic levels of MDA were presented in nmol/mL.

### 2.7. Determination of Plasma Antioxidant Capacity

The Trolox equivalent antioxidant capacity (TEAC) of plasma was performed, as previously described [25]. Briefly, a standard curve was established using 100 μg of Trolox (6-hidroxy-2,5,7,8-tetramethylchroman-2-carboxylic-acid; Sigma, St. Louis, MO, USA) in 1 mL of sodium acetate buffer (0.4 M, C_2_H_3_NaO_2_.3H_2_O) and glacial acetic acid (0.4 M). A plasma sample (20 μL) was added into a sodium acetate buffer and glacial acetic acid (200 μL) solution, and absorbance was read (at 660 nm) in a spectrophotometer (Synergy 4; BIOTEK, Winooski, VT, USA). Then, 20 μL of sodium acetate buffer (0.03 M) and glacial acetic acid (0.03 M) solution with H_2_O_2_ and ABTS (2,2′-azino-bis (3-ethylbenz-thiazolin-6 sulfonic acid; Sigma, St. Louis, MO, USA) was added to the samples and incubated for 5 min. Then, a second read (at 660 nm) in the spectrophotometer was performed. The second reading values were subtracted from the values found in the first reading, and the antioxidant activity of the sample was expressed as mmol of Trolox equivalent/L.

### 2.8. Zymography for MMP-2 Activity

Gelatin zymography method was performed in the placenta, as previously described [16,24,26]. Briefly, placenta samples were prepared using RIPA buffer (1 mM 1,10-ortho-phenanthroline, 1 mM phenylmethanesulfonyl fluoride), and 1 mM N-ethylmaleimide; (Sigma-Aldrich) containing protease inhibitor (4-(2-aminoethyl) benzenesulfonyl fluoride (AEBSF), E-64, bestatin, leupeptin, aprotinin, and EDTA). The samples were homogenized, and protein concentrations were measured by Bradford assay (Sigma-Aldrich). To separate the proteins, electrophoresis was performed using 12% acrylamide gels copolymerized with gelatin (0.05%), and 5μg of placenta proteins. The gels were washed twice for 30 min at room temperature in a Triton X-100 (2%) solution and incubated for 18 h in Tris–HCl buffer, containing 10 mmol/L CaCl_2_ at pH 7.4. Coomassie Brilliant Blue G-250 was used to stain the gels, and methanol solution was used for unstained gels. The gelatinolytic activity was measured using ImageJ software (1.43u version; NIH, Bethesda, MD, USA).

### 2.9. Data Analysis and Statistics

The results are expressed as mean ± SEM and analyzed in the GraphPad Prism^®^ software (version 8.0; San Diego, CA, USA). Shapiro–Wilk tests were applied to verify normality of data distribution. Repeated-measure analysis of variance (Two-away ANOVA) followed by Tukey post hoc tests were applied for multiple comparisons. A value of probability (*p*) <0.05 was considered statistically significant.

## 3. Results

### 3.1. Pravastatin Prevents Both Hypertensive Pregnancy and Placental Growth Restriction

Maternal systolic blood pressure on gestational day 19 showed values significantly reduced in the hypertensive pregnant rats treated with pravastatin as compared with untreated hypertensive pregnant rats (Figure 1 and Appendix A). The normotensive pregnant group treated with pravastatin presented systolic blood pressure measurements similar to those found in the untreated normotensive pregnant group (Figure 1 and Appendix A).

Following animal euthanasia and excising the uterus, the number of pups was not significantly different among the four groups (Figure 2 and Appendix A), suggesting no negative effect of pravastatin related to the fetuses.

To further confirm the effects of pravastatin on fetal–maternal interface, placental weights were assessed. The average placenta weight was only significantly reduced in untreated hypertensive pregnant rats compared to the other three groups (Figure 3 and Appendix A), indicating that pravastatin prevented placental growth restriction.

### 3.2. Pravastatin Prevents Both Reductions in Circulating Levels of NO Metabolites and Increases in Lipid Peroxide Levels in Plasma of Hypertension in Pregnancy, and Restores Plasma Antioxidant Capacity

Significant decreases in circulating NO metabolites in plasma were found in the HTN-Preg group compared to the Norm-Preg, Norm-Preg + Prava, and HTN-Preg + Prava groups (Figure 4 and Appendix A), suggesting that pravastatin prevented reductions in NO induced by hypertensive pregnancy.

Lipid peroxide levels in plasma were significantly increased in the HTN-Preg group compared to the Norm-Preg, Norm-Preg + Prava, and HTN-Preg + Prava groups (Figure 5 and Appendix A), demonstrating that pravastatin inhibited oxidative stress induced by hypertensive pregnancy.

Antioxidant capacity of plasma was significantly reduced in the HTN-Preg group compared to the Norm-Preg, Norm-Preg + Prava, and HTN-Preg + Prava groups (Figure 6 and Appendix A), demonstrating that pravastatin prevented the impairment in the plasma antioxidant capacity that induced hypertensive pregnancy.

### 3.3. Pravastatin Prevents Increases in Activity of MMP-2 in Placenta Induced by Hypertensive Pregnancy

Gelatin zymography analysis of placenta tissue homogenates revealed proteolytic bands corresponding to 75 Kda, 72 Kda, and 65 Kda MMP-2, while no bands corresponding to MMP-9 isoforms were observed (Figure 7A and Appendix A). The gelatinolytic activity of active MMP-2 isoform (65 Kda) in placenta was higher in the HTN-Preg group compared to Norm-Preg, Norm-Preg + Prava, and HTN-Preg + Prava groups (Figure 7D and Appendix A), while gelatinase activity of both inactive MMP-2 isoforms (75 Kda and 72 Kda) showed no differences among the four groups (Figure 7B and Figure 7C, respectively, and Appendix A).

### 3.4. Pravastatin Enhances Endothelium-Derived NO-Dependent Vasodilation

In endothelium-intact vessels, maximal acetylcholine-induced relaxation was similar in the aortas of all groups (Figure 8A and Appendix A). However, acetylcholine-induced vasodilation was enhanced in aortas of normotensive and hypertensive pregnant rats treated with pravastatin as compared with untreated rats (Figure 8A and Appendix A). Indeed, because the acetylcholine concentration-response curves were shifted to the left, acetylcholine was more potent in endothelium-intact aortas of groups treated with pravastatin compared with untreated groups (Figure 8A and Appendix A), suggesting the involvement of endothelial NO synthesis stimulated by pravastatin.

In endothelium-denuded aortas, acetylcholine-induced relaxation was completely inhibited in the aortas of all animal groups (Figure 8B), indicating that pravastatin effects are dependent on the endothelium.

In endothelium-intact aortas incubated with NO synthase inhibitor, L-NAME completely blocked acetylcholine-induced relaxations (Figure 8C), further supporting that pravastatin effects involved endothelium-derived NO.

## 4. Discussion

The main findings of the present study are that pravastatin: (I) attenuates hypertension and prevents placental weight loss; (II) prevents reductions in circulating NO metabolites; (III) protects against increases in oxidative stress and restores antioxidant capacity, (IV); mitigates increases in activity of MMP-2; and (V) enhances endothelium-derived NO-dependent vasodilation in rats with hypertension in pregnancy associated with placental weight loss.

Our current results are in line with previous studies showing that reductions in systolic blood pressure were also observed in rats with PE treated with pravastatin [4,5]. The antihypertensive effect of pravastatin is particularly important because elevation in maternal blood pressure is a concern in pre-eclamptic women [27]. Other negative outcomes of PE are the fetal and placental growth restrictions [4,28]. Concerning this, the present findings demonstrate no negative effect of pravastatin on fetuses, and pravastatin prevents placental weight loss, supporting the notion that there may be a favorable pharmacokinetic profile for pravastatin in PE [2,3,29,30,31]. In accordance with present results, previous clinical studies also showed that women with PE who were treated with pravastatin prolonged the days of pregnancy, and adverse neonatal outcomes were less common [32,33,34].

Here, we also observed that pravastatin restored circulating metabolites of NO in hypertensive pregnant rats, which may explain, at least in part, reductions in blood pressure and prevention of placental growth restriction in hypertensive pregnant rats treated with pravastatin. Corroborating our present results, earlier evidence has shown that pravastatin may induce NO synthesis in pre-eclamptic placentas by enhancing microsomal arginine uptake, as previously demonstrated [35], further indicating the importance of NO to underlie antihypertensive effects and to control the fetal–placental vascular tone, maintaining a high blood flow with low resistance circulation that favors placental development and oxygen and nutrient delivery to the fetus [2,4,11,27].

Because of oxidative stress, lipid peroxidation-derived bio-reactive agents impair NO formation in vascular diseases, including PE [4,36]. We also sought to determine products of lipid peroxidation and antioxidant capacity in maternal plasma. Increases in lipid peroxide levels and reductions in antioxidant capacity induced by hypertensive pregnancy were prevented by pravastatin. Indeed, increased oxidative stress is also found in women with PE [37] and rat models that have many of the features of PE [38]. As lipid peroxide levels are increased in response to oxidative imbalance, which is followed by impairment of antioxidants [38], our current results observed in pravastatin-treated hypertensive pregnant rats further support the idea that pravastatin has an antioxidant effect in the rat model of PE.

Oxidative stress has been shown to activate MMPs, modulating the activity through the disruption of zinc and residue of cysteine located in the catalytic site of the proenzymes [18]. Importantly, excessive increases in activity of MMPs have been associated with reductions in NO bioavailability and may contribute to the pathophysiology of hypertensive disorders of pregnancy, including PE [16,22,39]. These observations are in line with our current findings in which increases in activity of MMP-2 in the placenta and reduced NO bioavailability were found in untreated hypertensive pregnant rats, while pravastatin prevented increases in placental activity of MMP-2 and restored metabolites of NO. Corroborating previous studies, increases in activity of MMP-2 in placenta were also observed in the placenta of hypertensive pregnant rats [16] as well as in the plasma [40] and placenta of women with PE [17].

In general, it was reported that in healthy pregnancy there were increases in NO bioavailability with concomitant decreases in activity of MMP-2 in placenta of normotensive pregnant animals [16]. On the other hand, it was previously noted that under reduction in NO bioavailability there were increases in activity of MMP-2 in the placenta of hypertensive pregnant rats [16]. Our present results suggest that the disturbance of placental development due to maternal hypertension may be, at least in part, caused by increases in activity of MMP-2, while the antioxidant effect of pravastatin may have prevented placental growth restriction. However, further studies are warranted to confirm this observation.

To link the role of MMP-2 and oxidative stress with the blood pressure-lowering effect of pravastatin, it has been previously demonstrated that oxidative stress-induced reactive oxygen species, such as superoxide, may be involved in inactivation of NO as well as in activation of MMPs, thus promoting functional alterations of the cardiovascular system, and vascular remodeling in the hypertension secondary to imbalance activity of MMP-2 [41]. Superoxide reacts rapidly with NO thereby producing peroxynitrite, and a potential mechanism by which peroxynitrite leads to the activation of MMP-2 may involve the disruption in the binding between the zinc and the cysteinyl thiol in the auto-inhibitory pro-peptide domain of MMP-2, thus peroxynitrite exposes the catalytic site and activates MMP-2. Furthermore, it has been demonstrated that MMP-2-induced cleavage of adrenomedullin produces a vasoconstrictor out of a vasodilator [42], and MMP-2 cleaves big-endothelin-1 yielding a vasoconstrictor [43], which may explain, at least in part, increases in blood pressure observed in our present study. Therefore, in search of a potential antioxidant drug that may be directed towards the enhancement of endogenous NO and thereby attenuating both increases in maternal oxidative stress and activation of MMP-2, which in turn can reduce hypertension in PE and permit pregnancy to continue to term, pravastatin has revealed pleiotropic effects that include activation of endothelial NO synthase [2], reduction in placental oxidative stress [4,38], and increases in placental catalase activity [38], a key antioxidant enzyme that is activated in response to oxidative stress in the pre-eclamptic rat model [38]. This supports the present results of lipid peroxide levels in maternal plasma.

To support the idea that pravastatin effects involved endothelium-derived NO, acetylcholine-induced endothelium-dependent vasodilation was also assessed. Consistent with previous observations [4,44], pravastatin potentiated vasorelaxation responses to acetylcholine in aortic rings from normotensive and hypertensive pregnant rats treated with pravastatin, as compared with both untreated groups. Accordingly, underlying mechanisms were demonstrated to explain how pravastatin has enhanced the endothelium-derived NO dependent vasodilation. Previous studies have provided evidence that statins promote constitutive activation of endothelial NO synthase through several complementary mechanisms that include reduction in caveolin-1 and increasing heat shock protein 90, i.e., statins may act as a molecular chaperone, thereby facilitating long-term activation of endothelial NO synthase [1]. Other potential mechanisms relating to statins and NO may involve the activation of serine/threonine kinase Akt in endothelial cells, thus promoting the phosphorylation of endothelial NO synthase, which results in increases in NO synthesis [1]. Furthermore, these effects of pravastatin may also have contributed to the reductions in blood pressure and improvement of uteroplacental blood flow, corroborating with previous studies that reported the improvement of endothelial function in experimental models of PE [4,38,44]. Moreover, the involvement of NO in pravastatin treatment was confirmed by pre-incubation with a non-selective inhibitor of NO synthase (L-NAME) in abdominal aortic rings, in which L-NAME inhibited acetylcholine-induced endothelium-dependent relaxation, corroborating a previous study [4]. Taking together present and previous findings, endothelial dysfunction in PE may result from reduced NO bioavailability caused by increases in oxidative stress and may lead to the activation of MMPs, as previously suggested [4,16,38].

While the present study raises the possibility that pravastatin may protect against oxidative stress-induced activation of MMP-2 in the placenta, the current findings in rats should be carefully interpreted. Moreover, further studies are needed to investigate the cell-specific expression and activity of both MMP-2 and MMP-9 in the placenta, since during normal pregnancy the placental basal villi are tightly attached to the decidual tissue to anchor the placenta within the uterus, and this attachment depends on cell–cell and cell–extracellular matrix interaction [45], while this process in PE is still unclear.

## 5. Conclusions

This study provides evidence that pravastatin attenuates both increases in systolic blood pressure and placental weight loss induced by experimental modeling of PE, and probably the antioxidant effects of pravastatin protect against oxidative stress-induced activation of MMP-2 as well as leading to the enhancement of endothelium-derived NO-dependent vasodilation. Our findings suggest that pravastatin may be a promising drug for prevention or therapy of the changes caused by PE.

## Figures and Tables

**Figure 1 antioxidants-12-00939-f001:**
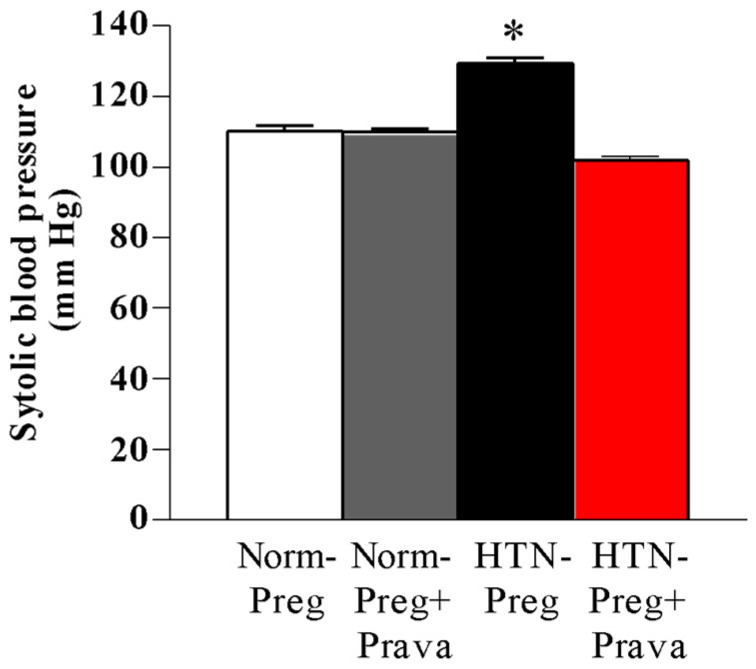
Systolic blood pressure measured in Norm-Preg, Norm-Preg + Prava, HTN-Preg, and HTN-Preg + Prava groups. Values represent mean ± SEM. * *p* < 0.05 vs. Norm-Preg group.

**Figure 2 antioxidants-12-00939-f002:**
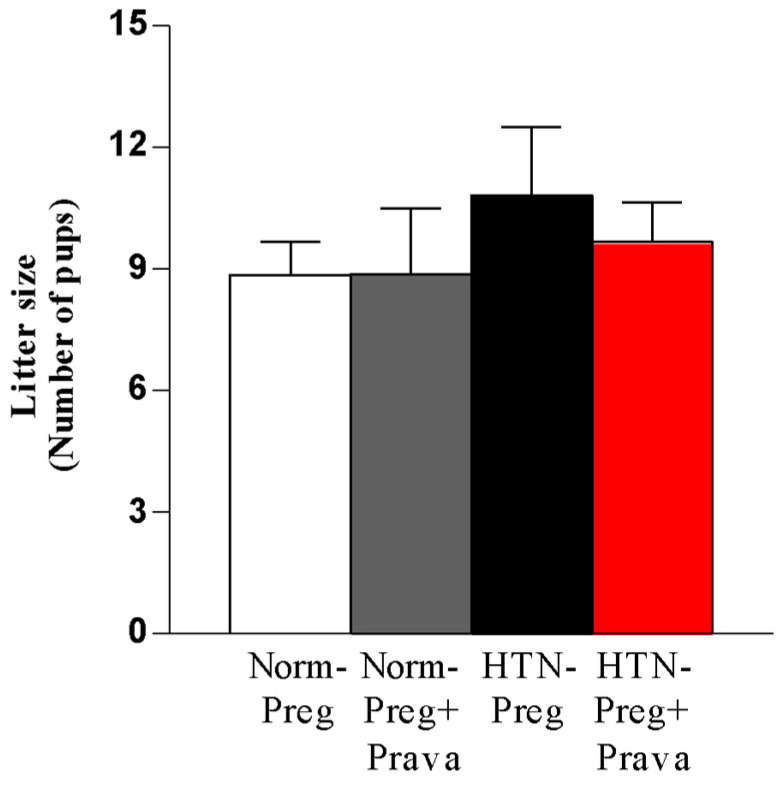
Litter size recorded in Norm-Preg, Norm-Preg + Prava, HTN-Preg, and HTN-Preg + Prava groups. Values represent mean ± SEM.

**Figure 3 antioxidants-12-00939-f003:**
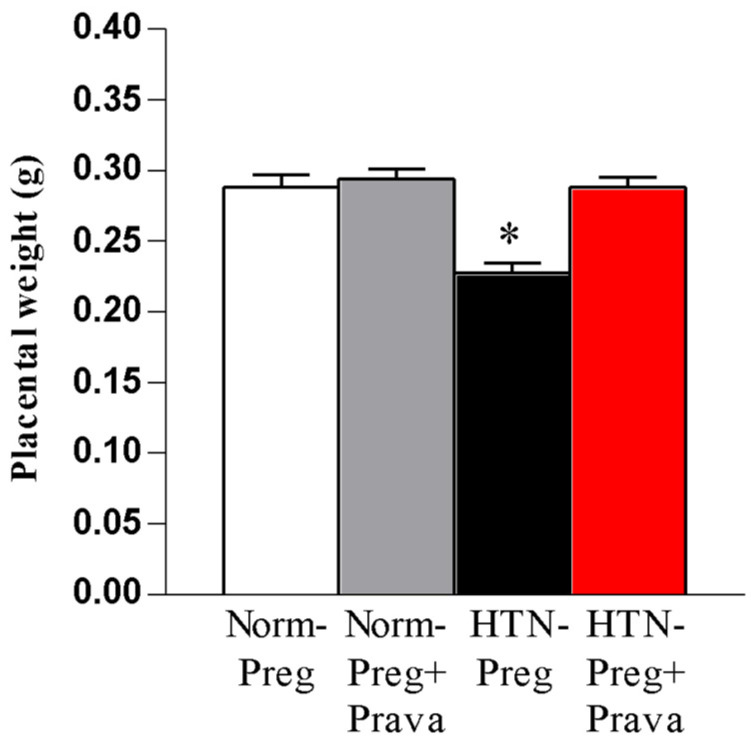
Placental weights recorded in Norm-Preg, Norm-Preg + Prava, HTN-Preg, and HTN-Preg + Prava groups. Values represent mean ± SEM. * *p* < 0.05 vs. Norm-Preg group.

**Figure 4 antioxidants-12-00939-f004:**
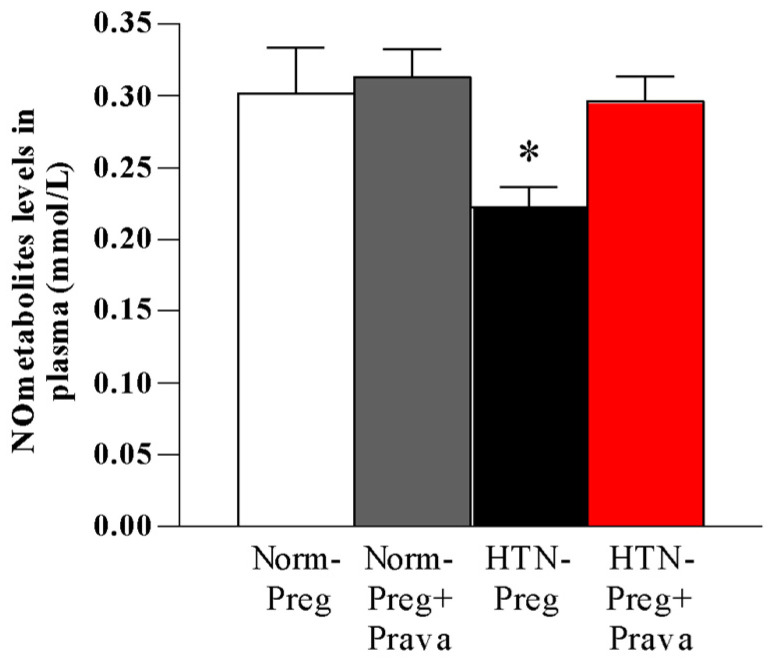
Plasma levels of NO metabolites determined in Norm-Preg, Norm-Preg + Prava, HTN-Preg, and HTN-Preg + Prava groups. Values represent mean ± SEM. * *p* < 0.05 vs. Norm-Preg group.

**Figure 5 antioxidants-12-00939-f005:**
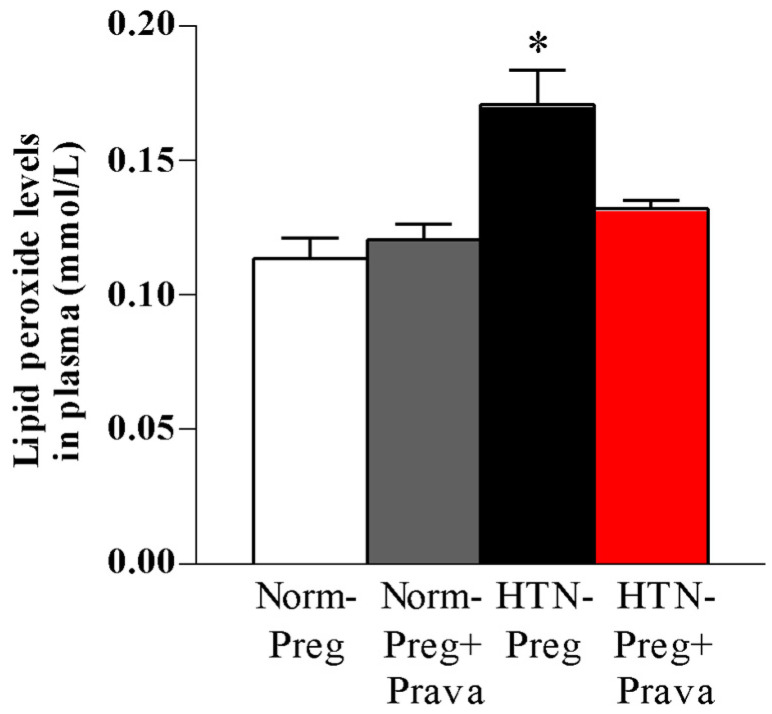
Lipid peroxide levels in plasma determined in Norm-Preg, Norm-Preg + Prava, HTN-Preg, and HTN-Preg + Prava groups. Values represent mean ± SEM. * *p* < 0.05 vs. Norm-Preg group.

**Figure 6 antioxidants-12-00939-f006:**
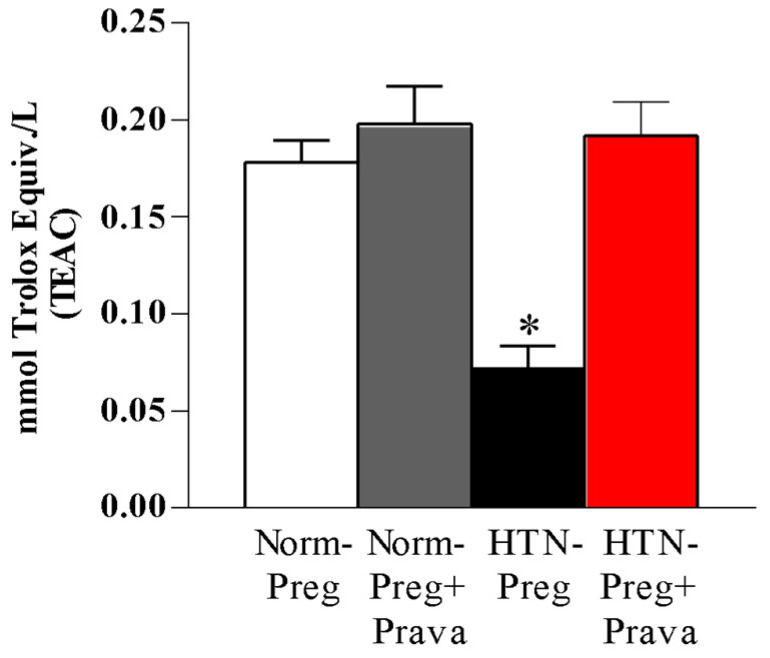
The Trolox equivalent antioxidant capacity (TEAC) in plasma determined in Norm-Preg, Norm-Preg + Prava, HTN-Preg, and HTN-Preg + Prava groups. Values represent mean ± SEM. * *p* < 0.05 vs. Norm-Preg group.

**Figure 7 antioxidants-12-00939-f007:**
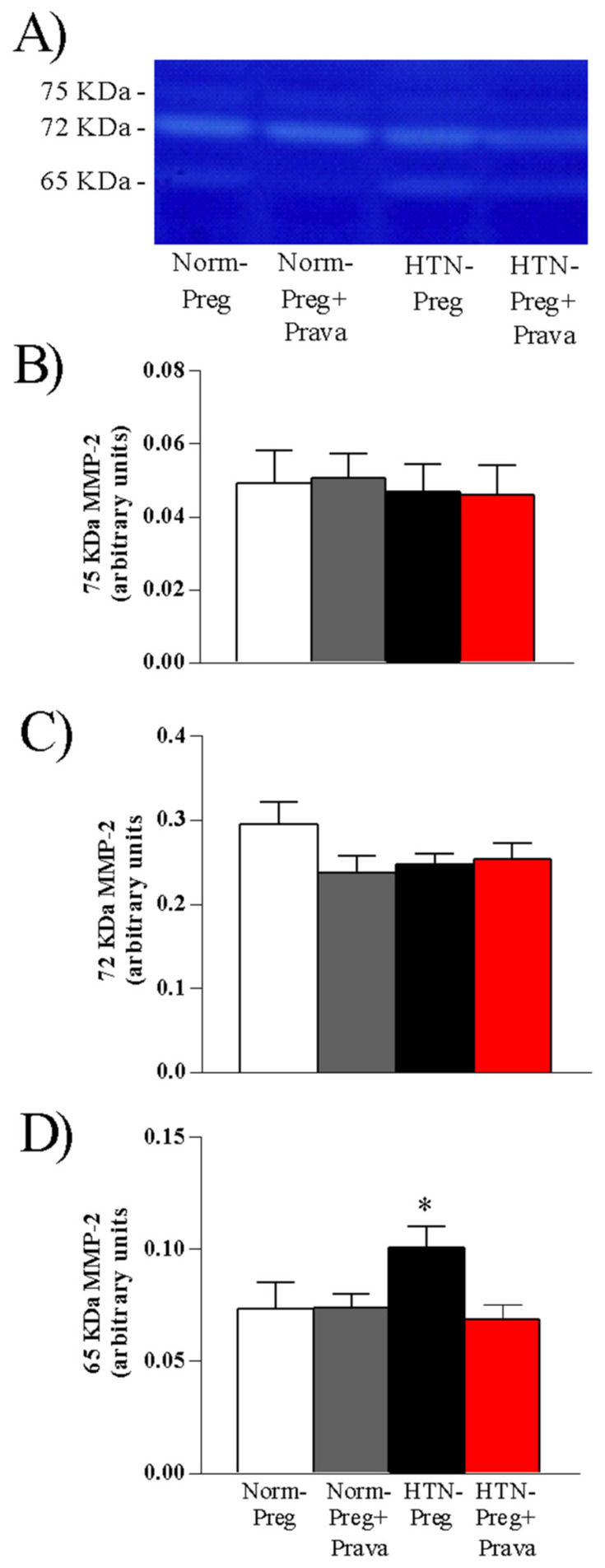
Representative zymography gel of placenta (**A**), gelatinolytic activities of 75 Kda MMP-2 (**B**), 72 Kda MMP-2 (**C**), and 65 KDa MMP-2 (**D**) of Norm-Preg, Norm-Preg + Prava, HTN-Preg, and HTN-Preg + Prava groups. Values represent mean ± SEM. * *p* < 0.05 vs. Norm-Preg group.

**Figure 8 antioxidants-12-00939-f008:**
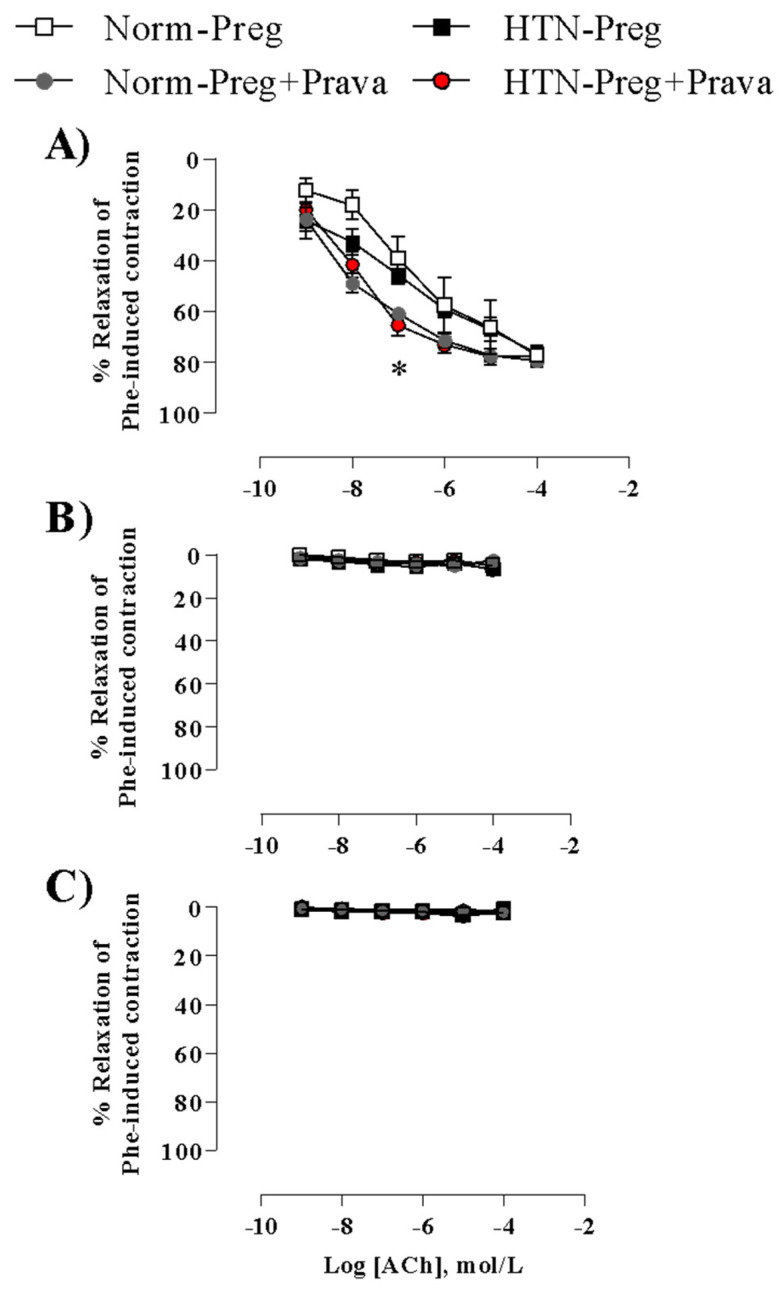
Acetylcholine (ACh)–induced relaxation (%) after precontraction with phenylephrine (Phe) in aortas with endothelium (**A**), aortas without endothelium (**B**), and aortas with endothelium and incubated with NO synthase inhibitor, L-NAME (**C**) of Norm-Preg, Norm-Preg + Prava, HTN-Preg, and HTN-Preg + Prava groups. Values represent mean ± SEM. * *p* < 0.05 vs. Norm-Preg and HTN-Preg.

## Data Availability

Authors declare that all the data supporting the results of the present study are included in the article.

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
