# Peer review of "Pravastatin Prevents Increases in Activity of Metalloproteinase-2 and Oxidative Stress, and Enhances Endothelium-Derived Nitric Oxide-Dependent Vasodilation in Gestational Hypertension"

_antioxidants, 2023, doi:10.3390/antiox12040939_

Round 1

Reviewer 1 Report

This study investigated the potential effects of pravastatin, a cholesterol-lowering medication, on blood pressure, placental weight, and oxidative stress in an experimental preeclampsia (PE) model. Overall, the study provides promising evidence for the potential use of pravastatin as a therapeutic intervention for PE, with minor changes needed to improve the manuscript.

1) The first time that uses NO, indicates that is nitric oxide.

2) Although the graphics are good, please provide tables with the data of the graphics. This will be done the paper more friendly to those readers that want to perform in the future either a systematic review or a Meta-analysis.

Author Response

Reviewer 1 #

This study investigated the potential effects of pravastatin, a cholesterol-lowering medication, on blood pressure, placental weight, and oxidative stress in an experimental preeclampsia (PE) model. Overall, the study provides promising evidence for the potential use of pravastatin as a therapeutic intervention for PE, with minor changes needed to improve the manuscript.

We thank you for your careful revision, helpful comments and suggestions. Please find below our responses.

1) The first time that uses NO, indicates that is nitric oxide.

We have reviewed all acronyms in the manuscript to ensure that prior to first use they are spelled out including nitric oxide (NO).

2) Although the graphics are good, please provide tables with the data of the graphics. This will be done the paper more friendly to those readers that want to perform in the future either a systematic review or a Meta-analysis.

            We totally agree with you. We have now provided a supplementary table with all data of the graphs.

Reviewer 2 Report

The sent article entitle: Pravastatin Prevents Increases in Activity of Metalloproteinase-2 and Oxidative Stress, and Enhances Endothelium-derived Nitric oxide-dependent Vasodilation in Gestational Hypertension has taken into the importance from the pregnancy/neonatology point of view hypertension. Therefore, the link between oxidative stress and pravastatin's protecting role is interesting and worth investigating. The article has been well written and readable, the experimental part is clear and easy to follow. In the Abstract and Conclusion parts Authors should consider the medical benefit of their observation. In the Introduction, I observed the lack of NO secretion biochemistry as well as the numbers of preeclampsia/hypertension cases and its depend (if any) on countries developed low and high. It would be beneficial if the Authors put some information about nutrition/feeding influence on PE. 

However, in my opinion, the article is out of the Antioxidant journal scope. I strongly recommend authors send the manuscript to a more specialistic medical periodic.

Author Response

Reviewer 2 #

The sent article entitle: Pravastatin Prevents Increases in Activity of Metalloproteinase-2 and Oxidative Stress, and Enhances Endothelium-derived Nitric oxide-dependent Vasodilation in Gestational Hypertension has taken into the importance from the pregnancy/neonatology point of view hypertension. Therefore, the link between oxidative stress and pravastatin's protecting role is interesting and worth investigating. The article has been well written and readable, the experimental part is clear and easy to follow.

We thank you for your careful revision, helpful comments and suggestions. Please find below our responses.

In the Abstract and Conclusion parts Authors should consider the medical benefit of their observation.

            We agree with you. We have now included a statement in the Abstract and Conclusion of the revised version of the manuscript as follow:

Abstract:

“…, thus suggesting pravastatin as a therapeutic intervention for PE.”

Conclusion:

“…. Our findings suggest that pravastatin may be a promising drug for prevention or therapy of the changes caused by PE.”

In the Introduction, I observed the lack of NO secretion biochemistry as well as the numbers of preeclampsia/hypertension cases and its depend (if any) on countries developed low and high. It would be beneficial if the Authors put some information about nutrition/feeding influence on PE.

            We agree with you that Introduction section needs more biochemistry evidence related to NO. Thus we have added it as follow:

“….It has been demonstrated that constitutive activity of endothelial NO synthase is directly enhanced by statins, which underlying mechanisms may involve reduction of caveolin-1 and concomitant increase in heat shock protein 90, thus facilitating long-term activation of endothelial NO synthase, thereby increasing endogenous NO synthesis [1].”

            We have also added in the Introduction section statements related to number of cases of preeclampsia in developing and developed countries as well as information about nutrition/diet influences on PE, as follow:

“PE incidence in developing countries is 1.8-16.7%, while about 0.4% of incidence is reported in developed countries [6]. Development of PE has shown influences that include environmental and genetic factors, and lifestyle [7]. Although little information is known relating nutrition/diet and PE, it has been reported that 10% of pregnant obese women develop PE and that these two conditions share common pathophysiological mechanisms [8], thereby nutritional guidelines to reduce obesity in pregnancy are of particular importance to attenuate risk of PE [9].”

However, in my opinion, the article is out of the Antioxidant journal scope. I strongly recommend authors send the manuscript to a more specialistic medical periodic.

            We respectfully have to disagree with you. Please see our revised version in which additional data related to antioxidant capacity of pravastatin were added in the manuscript. Please see the section 2.7 in Methods, the new Figure 6 in the section 3.2 in Results, and respective statements highlighted in red in the Discussion section.

Reviewer 3 Report

1. The manuscript is more descriptive without molecular mechanism insight.

2. A graphic abstract for mechanistic pathway of pravastatin is highly recommended.

3. A biostatistician is highly recommended for this study.

4. Why only MMP-2 activity was examined? The expression and activity of MMP-2 and MMP-9 should be examined. The whole image of zymography assay is required for this study.

5. Does MMP activity affect NO production and blood pressure? The experimental evidence is required for supporting how to link the role of MMPs and oxidative stress to the blood pressure-lowering effect of pravastatin.

6. What are the NO metabolites?

7. The histological examination of placenta in 4 groups are required for this study.

8. The lipid peroxide levels in placenta should be examined.

9. What’s the effects of pravastatin on the capacity of antioxidants? This comment should be experimentally addressed. This evidence is important for the publication of Antioxidants.

10. Can gestational Hypertension induce systemic or local (placenta) inflammation?

Author Response

Reviewer 3 #

  1. The manuscript is more descriptive without molecular mechanism insight.

            We thank you for your careful review, but we partially agree with you. Importantly, our present study is the first to investigate the effects of pravastatin on the gelatinolytic activities of inactive isoforms of MMP-2 (75 KDa and 72 KDa, respectively) as well as on the gelatinolytic activity of active isoform of MMP-2 (65 KDa) in a rat model of PE, and that provides evidence that pravastatin protects against activation of MMP-2 induced by oxidative stress in preeclamptic rats. Our present results may also involve improvement of endothelial function and antihypertensive effects of pravastatin that may be related to NO. Also, our present study provides evidence for the potential use of pravastatin as a therapeutic intervention for PE. Please see our revised version in which additional data related to antioxidant capacity of pravastatin were added in the manuscript.

  1. A graphic abstract for mechanistic pathway of pravastatin is highly recommended.

            We totally agree with you. A graphical abstract is now provided.

  1. A biostatistician is highly recommended for this study.

            We asked a biostatistician to revise all statistical analyses as well as the text of the respective section. Please find below our revised section in the revised version of the manuscript.

2.9. Data analysis and statistics

The results are expressed as mean ± SEM and analyzed in the GraphPad Prism® software (version 8.0, San Diego, CA, USA). Shapiro–Wilk tests were applied to verify normality of data distribution. Repeated-measure analysis of variance (Two-away ANOVA) followed by Tukey post-hoc tests were applied for multiple comparisons. A value of probability (P) <0.05 was considered statistically significant.

  1. Why only MMP-2 activity was examined? The expression and activity of MMP-2 and MMP-9 should be examined.

Because we have performed gelatinolytic activity assays with placenta before the term in pregnancy in rats, and, in our hands, corresponding bands to MMP-9 isoforms were not detected in our study. Supporting this, MMP-2 has been detected around the blood vessels, while MMP-9 has not been detected in blood vessels. Furthermore, in accordance with our results, a previous study demonstrated that both MMP-2 and MMP-9 exhibit time (labor and nonlabor fetal) and cell-specific expression in the human placenta [45]. Authors provided evidence that MMP-2 was mainly detected around the placental vessels, while there was very weak staining for MMP-9 in placental syncytiotrophoblasts from placental villi [45]. Moreover, some individual cells of the placental segments that lack MMP-9 showed positive MMP-2 staining [45]. The cell-specific expression and gelatinolytic activities of MMP-2 and MMP-9 may be accounted for by the different matrix distribution in the tissues of maternal-fetal interface, and the substrate difference for these two enzymes, as previously suggested [45]. However, further studies need to investigate the cell-specific expression and activity of MMP-2 and MMP-9 in placenta, since during normal pregnancy, the placental basal villi are tightly attached to the decidual tissue to anchor the placenta within the uterus, and this attachment depends on cell-cell and cell-extracellular matrix interaction, while this process in PE is still unclear.

We have included these statements as limitations of the study, as follow:

“While the present study raises the possibility of pravastatin may protect against oxidative stress-induced activation of MMP-2 in placenta, the current findings in rats should be carefully interpreted, and, further studies need to investigate the cell-specific expression and activity of both MMP-2 and MMP-9 in placenta, since during normal pregnancy, the placental basal villi are tightly attached to the decidual tissue to anchor the placenta within the uterus, and this attachment depends on cell-cell and cell-extracellular matrix interaction [45 ], while this process in PE is still unclear.”

The whole image of zymography assay is required for this study.

The intact (whole image) original gels of Zymography of placenta are now attached to this study as supplementary figure.

  1. Does MMP activity affect NO production and blood pressure? The experimental evidence is required for supporting how to link the role of MMPs and oxidative stress to the blood pressure-lowering effect of pravastatin.

            A suggested mechanism has been proposed as follow:

Oxidative stress-induced reactive oxygen species may be involved in inactivation of NO as well as in activation of MMPs, thus promoting functional alterations of cardiovascular system, and vascular remodeling secondary to imbalance activity of MMP-2 in the hypertension. In addition, MMP-2 may cleavage key peptides that are involved in controlling of blood pressure.

Please see our new sentences below that are added in the Discussion section.

“To link the role of MMP-2 and oxidative stress with the blood pressure-lowering effect of pravastatin, it has been previously demonstrated that oxidative stress-induced reactive oxygen species, such as superoxide, may be involved in inactivation of NO as well as in activation of MMPs, thus promoting functional alterations of cardiovascular system, and vascular remodeling in the hypertension secondary to imbalance activity of MMP-2 (41). Superoxide reacts rapidly with NO thereby producing peroxynitrite, and a potential mechanism by which peroxynitrite lead to the activation of MMP-2 may involve the disruption of the binding between the zinc and the cysteinyl thiol in the auto-inhibitory pro-peptide domain of MMP-2, thus peroxynitrite exposes the catalytic site and activates MMP-2. Furthermore, it has been demonstrated that MMP-2-induced cleavage of adrenomedullin produces a vasoconstrictor out of a vasodilator [42], and MMP-2 cleaves big-endothelin-1 yielding a vasoconstrictor [43], which may explain, at least in part, increases in blood pressure observed in our present study. Therefore, in search of potential antioxidant drug that may be directed toward the enhancement of endogenous NO and thereby attenuating both increases in maternal oxidative stress and activation of MMP-2, which in turn can reduce hypertension in PE and permit pregnancy to continue to term. “

  1. What are the NO metabolites?

            The NO metabolites are nitrite and nitrate. Please see section 2.5 in the revised version of the manuscript.           

  1. The histological examination of placenta in 4 groups are required for this study.

            Unfortunately, placentas are over, since all placenta samples were used to quantify gelatinolytic activity of MMP-2 isoforms. Therefore, we included in Discussion section the follow sentences:

“… further studies need to investigate the cell-specific expression and activity of both MMP-2 and MMP-9 in placenta, since during normal pregnancy, the placental basal villi are tightly attached to the decidual tissue to anchor the placenta within the uterus, and this attachment depends on cell-cell and cell-extracellular matrix interaction [45], while this process in PE is still unclear.”

  1. The lipid peroxide levels in placenta should be examined.

            As answered to your previous question (Comment number 7), all placentas were used in zymography gels. However, we added the following sentence in the Discussion section to support our results relating anti-oxidative stress effects of pravastatin in maternal plasma that are corroborated by previous studies demonstrating both reductions in placental oxidative stress and increases in placental activity of antioxidant enzyme were found in preeclamptic rats treated with pravastatin, as follow.

“……. Pravastatin has revealed pleiotropic effects that includes activation of endothelial NO synthase [2], reduction of placental oxidative stress [4,38] and increases in placental catalase activity [38], a key antioxidant enzyme, that is activated in response to oxidative stress in preeclamptic rat model [38], thus supporting the present results ….”

  1. What’s the effects of pravastatin on the capacity of antioxidants? This comment should be experimentally addressed. This evidence is important for the publication of Antioxidants.

            We agree with you. Fortunately, we had still plasma samples of this project, thus to examine effects of pravastatin on the capacity of antioxidants, we have experimentally addressed the total antioxidant capacity in plasma, in which Trolox-equivalent antioxidant capacity of the plasma was determined. Please see the section 2.7 in Methods, the new Figure 6 in the section 3.2 in Results, and respective statements highlighted in red in the Discussion section.

  1. Can gestational Hypertension induce systemic or local (placenta) inflammation?

            Both systemic and local (placental) inflammation may occur in women with PE. It has been demonstrated that preeclamptic women have placental inflammation with concomitant autoantibodies production. One of the main theories related to the causes of PE is placental ischemia, which is resulted of inefficient trophoblast invasion to the maternal endometrium, due to immune imbalance in which pro-inflammatory cells are increased and T regulatory cells are decreased. Further, preeclamptic placenta releases various bioactive factors including pro-inflammatory cytokines into maternal circulation, and that are associated with systemic inflammation followed by oxidative stress, as previously revised [https://www.ncbi.nlm.nih.gov/pmc/articles/PMC5484393/ ].

            Please see our statement in Introduction as follow:

“The hypoxic environment in preeclamptic placenta results in release of reactive oxygen species (ROS) and endothelial dysfunction mediators, such as lipid peroxides and pro-inflammatory cytokines, which impair NO bioavailability [10-12].  Accordingly …”

Round 2

Reviewer 2 Report

The authors carried out the requested changes and the manucript can be accepted.

Reviewer 3 Report

Authors have addressed my comments.